# Dual-Function Meta-Grating Based on Tunable Fano Resonance for Reflective Filter and Sensor Applications

**DOI:** 10.3390/s23146462

**Published:** 2023-07-17

**Authors:** Feifei Liu, Haoyu Jia, Yuxue Chen, Xiaoai Luo, Meidong Huang, Meng Wang, Xinping Zhang

**Affiliations:** 1College Physics & Materials Science, Tianjin Normal University, Tianjin 300387, China; 2210300023@stu.tjnu.edu.cn (H.J.); 15075451840@163.com (Y.C.); luoxiaoai0304@163.com (X.L.); mdhuang@tjnu.edu.cn (M.H.); 2School of Physical Science and Technology, Inner Mongolia University, Hohhot 010021, China; wangmeng@imu.edu.cn; 3Institute of Information Photonics Technology, College of Applied Sciences, Beijng University of Technology, Beijing 100124, China; zhangxinping@bjut.edu.cn

**Keywords:** localized surface plasmon resonance, Rayleigh anomaly diffraction, Fano resonance, reflective filter, sensor

## Abstract

Localized surface plasmon resonance (LSPR)-based sensors exhibit enormous potential in the areas of medical diagnosis, food safety regulation and environmental monitoring. However, the broadband spectral lineshape of LSPR hampers the observation of wavelength shifts in sensing processes, thus preventing its widespread applications in sensors. Here, we describe an improved plasmonic sensor based on Fano resonances between LSPR and the Rayleigh anomaly (RA) in a metal–insulator–metal (MIM) meta-grating, which is composed of silver nanoshell array, an isolation grating mask and a continuous gold film. The MIM configuration offers more freedom to control the optical properties of LSPR, RA and the Fano resonance between them. Strong couplings between LSPR and RA formed a series of narrowband reflection peaks (with a linewidth of ~20 nm in full width at half maximum (FWHM) and a reflectivity nearing 100%) within an LSPR-based broadband extinction window in the experiment, making the meta-grating promising for applications of high-efficiency reflective filters. A Fano resonance that is well optimized between LSPR and RA by carefully adjusting the angles of incident light can switch such a nano-device to an improved biological/chemical sensor with a figure of merit (FOM) larger than 57 and capability of detecting the local refractive index changes caused by the bonding of target molecules on the surface of the nano-device. The figure of merit of the hybrid sensor in the detection of target molecules is 6 and 15 times higher than that of the simple RA- and LSPR-based sensors, respectively.

## 1. Introduction

Localized surface plasmon resonances, as a result of the collective oscillation of free electrons in noble metals, show great potential in chemical/biological sensors owing to their capability of manipulating electromagnetic waves in the nanoscale and inherent sensitivity to changes in refractive index (RI) of the local environment [1,2,3,4,5]. However, the figure of merit (FOM) value, which is defined as the value of RI sensitivities divided by resonance linewidths in the FWHM of LSPR sensors [6], is generally small due to the broadband spectral lineshape of LSPR (FOM < 15) [7,8,9,10,11], thus preventing the widespread application of LSPR sensors. To solve this problem, an effective method is to introduce one more resonance mode with a high quality factor (Q), such as the waveguide mode [12,13], optical microcavity resonance mode [14,15], high-order plasmonic resonance [16,17,18] or Rayleigh anomaly diffraction mode [19,20,21], to the system and then utilize the newly generated Fano resonance between the LSPR and the high-Q modes for sensor applications [22]. The hybridization between LSPR and high-Q resonance modes not only reduces the linewidths of LSPR-based sensors, but also modulates the enhanced electric-intensity distribution near the nanostructure [23,24,25,26], making it more accessible and sensitive to molecular species. Therefore, the FOM value of an LSPR sensor can be effectively increased by such mode-coupling processes [27]. 

In the above-mentioned works, a hybrid sensor based on Fano resonance between LSPR and RA is worthy of attention. The Rayleigh anomaly, corresponding to a diffraction tangential to the grating surface, is well known for its sharply asymmetric lineshape [28] and high sensitivity to changes in the RI of the environment (with the sensitivity proportional to the period of grating) [29,30,31]. Those exotic characteristics make RA-related hybrid sensors more suitable for the detection of low-concentration biological/chemical molecules or explosive molecules [11,19]. Unfortunately, in the reported works, due to a lack of suitable platforms for carefully regulating the coupling relationship between LSPR and RA to modulate the enhanced electric-intensity distribution near the nanostructure, the abilities of this kind of Fano resonance in measuring local RI changes caused by the adsorption of target molecules on the surface of a nano-device are not yet fully exploited. 

In this work, we investigate a tunable Fano resonance between LSPR and RA in an MIM meta-grating composed of a silver nanoshell array, an isolation grating mask and a continuous gold film. The MIM configuration offers more degrees of freedom to control the optical properties of LSPR, RA and the Fano resonance between them. Both experimental and simulated results show that the coupling behavior between LSPR and RA including coupling strength and spectral contrast can be modulated effectively by the angles of incident light in an optimized meta-grating. Strong couplings between LSPR and RA worked at large incident angles form a series of narrowband and high-contrast reflective peaks (with a linewidth of ~20 nm in the FWHM and a reflectivity of closing to 100%) in an LSPR-based dark background and are suitable for the applications of high-efficiency reflective filters. A Fano resonance that is well optimized by carefully adjusting the angles of incident light can enhance effectively the light–matter interaction in the nanoscale and thus switch such a nano-device to a high-performance biological/chemical sensor with an FOM larger than 57 and the capability to detect the binding of small (organic) molecules on the nano-device surface. The FOM of the hybrid sensor in the detection of local RI changes is 6 and 15 times higher than that of simple RA- and LSPR-based sensors, respectively.

## 2. Results

### 2.1. Design and Characterization of the MIM Meta-Grating

Figure 1a schematically depicts the geometry of the meta-grating we designed, which is composed of an array of silver nanoshells and a continuous gold film separated by a photoresist (PR) grating mask that is not developed to the end. The gold film prohibits any transmission through the system so that only the reflection of the device needs to be considered, laying the basis for generating high-contrast RA diffraction [32]. Meanwhile, the gold film is relatively stable in air and chemical media, which provides convenience for the long-process parameter optimization on the MIM meta-grating. Near-field coupling between two metallic layers (i.e., Ag nanoshells and the gold film) forms a series of LSPRs [33,34,35,36] in different wavelengths, with their features dictated by geometrical details as defined in Figure 1a. The structural parameters of the MIM meta-grating are defined as follows: the thickness of the Au film *d*_1_; the thickness of the photoresist spacer *d*_2_; the modulation depth of the PR grating *h*; the full width at half maximum of the PR grating lines *W*; the thickness of the Ag nanoshells *t*; the period of the nanocavity array *P*. For the MIM meta-grating, Ag was chosen as the metallic material of nanoshells for its relatively low ohmic loss, which is helpful for improving the performance of LSPR-based sensors. Figure 1b,c show the fabrication schemes of the MIM meta-grating. Firstly, interference lithography was employed to fabricate the PR grating mask on the surface of a silica substrate coated with a 70 nm thickness Au film, where a He-Cd laser at 325 nm and photoresist S1805 were applied as the UV light source and the recording medium, respectively. Then, Ag nanoshells were evaporated on the surface of the PR grating using a two-step oblique thermal evaporation method, as shown in Figure 1c, where the oblique evaporation angle 𝛼 is related to the structural parameters of full width at half maximum of the PR grating lines *W*, the modulation depth *h* and the period of the PR grating *P*. To obtain a high-quality metallic film, we chose an evaporation speed of about 1.5 nm/s in the experiment. Figure 1b,c depict, respectively, the scanning electron microscopy (SEM) pictures of part of the fabricated sample before and after the deposition of Ag nanoshells on the surface of the PR grating. Through parameter sweeps on the geometrical parameters, such as the full width at half maximum of the PR grating lines and the thickness of the photoresist spacer and Ag nanoshells, using finite-difference time-domain (FDTD) simulations, we optimized the geometric parameters of the MIM meta-grating in the experiments as follows: *P* ≈ 465 nm, *d*_1_ ≈ 70 nm, *d*_2_ ≈ 50 nm, *h* ≈ 260 nm, *W* ≈ 165 nm, *t* ≈ 50 nm and oblique evaporation angle α = 50°.

We experimentally characterized the optical properties of the fabricated samples depicted in Figure 1d,e for the sake of comparison. Red circles in Figure 2a,b represent the measured reflectance spectra of the samples under the illumination of normally incident light for TM polarization (perpendicular to the extending direction of grating lines), which are in nice agreement with corresponding FDTD simulations as plotted by blue solid curves in Figure 2a,b, respectively. In our simulations, geometric parameters of the simulated nanostructures were derived from the experiments. Meanwhile, the intrinsic absorptions of gold were also considered. For the dielectric grating without Ag nanoshells, the RA resonances in two spectra (i.e., the measured and simulated reflectance spectra of the grating) as marked by black dotted circular rings in Figure 2a are observed as two extremely shallow dips at about 463 nm due to their natural sub-radiation properties. The linewidth of RA modes in the FWHM was about 8 nm in the experiment. The other two dips appearing at 645 nm and 740 nm in two spectra correspond to surface plasmon polaritons (SPPs), which are bound to the interface between the Ag nanoshell grating and Au film and the interface between the Au film and the SiO_2_ substrate, respectively (see Appendix A in the Appendix A for the corresponding electric-intensity distribution). The resonance wavelength of these two SPPs at a normal incident angle can be written as $\lambda_{SPP}=N_{eff}^{G}P$, where P is the grating period and $N_{eff}^{G}$ is the effective refractive index for the Ag nanoshell grating–Au film regime (this regime is related to the duty cycle of PR grating) or SiO_2_ substrate–Au film regime [37]. Those spectral features can also be confirmed by the corresponding angle-resolved reflectance spectrum below.

As to the MIM meta-grating, the RA resonances in two spectra as highlighted by the black and red dotted circular rings, respectively, in Figure 2b turn to weak peaks due to the relatively weak Fano couplings between RA and the broadband plasmonic modes with their central wavelength at λ≈610 nm. Near-field interaction between Ag nanoshells and gold film forms such broadband plasmonic modes with stronger absorption and electric-field enhancement (~150 nm in the FWHM and a reflectivity below 20% at central wavelength), which lays the basis for the realization of tunable Fano resonances in a wide range. Compared to the simple RA mode mentioned in Figure 2a, the linewidth of the hybrid RA modes in the FWHM was increased to 12 nm in the experiment. To quantitatively evaluate the optical properties of RA, the plasmonic modes and the coupling behaviors between them, we apply the coupled-mode theory (CMT) mentioned in [16,34] to help us precisely extract the basic parameters of the resonance modes, including the intrinsic damping, the radiation damping, and the coupling strength between these two resonance modes, where the system is simplified as a one-port (reflection geometry) two-mode (considering only the RA and LSPR modes in this work) model. The black solid curve in Figure 2b shows the CMT fitting spectrum with parameters as follows: radiation damping of RA and the broadband LSPR: *Γ_RAe_* = 5, *Γ_LSPRe_* = 25; intrinsic damping of these two modes: *Γ_RAi_* = 0.5, *Γ_LSPRi_* = 55; coupling strength between these two resonance modes: *t* = 5. Good agreement is observed between the measurement and the CMT fitting. In this case (i.e., θ=0°), the far-field radiation ability and the linewidth of the hybrid RA mode are slightly influenced by the relatively weak coupling process.

It is worth noting that the relatively low reflectance of the MIM meta-grating in the wavelength range of 400–550 nm actually results from the intrinsic absorption of the gold film, which can be verified by the measured reflectance spectra of the gold film as depicted in Figure 2c and plotted by blue circles. Therefore, the spectral superposition of the intrinsic absorption of the gold film and the broadband LSPR at λ≈610 nm can perfectly reproduce the relatively complex spectral lineshape of the MIM meta-grating in Figure 2b, which can be seen from the CMT fitting results as depicted in Figure 2c and plotted by blue (the intrinsic absorption of the gold film) and red (the broadband LSPR) solid curves, respectively. Meanwhile, the reflectance peaks appear at 720 nm, corresponding to the Fano resonances between the broadband LSPR and the SPP mode (see Appendix A in the Appendix A for the corresponding electric-intensity distribution), which can also be demonstrated by the following angle-resolved reflectance spectra. Figure 2d shows the electric-intensity distribution of the MIM meta-grating at λ≈465 nm and 610 nm, corresponding to the RA and LSPR modes, respectively, where the enhanced electric fields are mainly distributed on the surface of the MIM grating and the interspace between two Ag nanoshells, respectively. Appendix A shows the current and charge-density distribution of the MIM meta-grating at these two resonance modes. Due to the weak Fano coupling between LSPR and RA, the near-field distribution of RA is slightly modulated by the LSPR, resulting in a relatively weak electric field localized in the interspace between two Ag nanoshells.

Figure 3a,b show the angle-resolved reflectance spectra of the fabricated samples depicted in Figure 1d,e, respectively, with the incident angles turned from 0° to 10°, where the spectral shift of RA resonances is highlighted by a series of dotted circular rings. Three obvious features near the RA diffraction wavelengths can be observed: (1) The diffracted wavelengths of RA in experiments are basically consistent with the theoretical calculations as plotted by black dotted lines in Figure 3a,b. The central wavelength of first-order RA is determined by the diffraction formula of P(sin(θ)+n)=λ  [31], where n is the refractive index of the environment and equals 1 when the grating is located in air, P is the grating period, and λ is the diffracted wavelength of RA. Such an angle-resolved behavior further verifies the identities of RA diffractions. (2) The RA resonances in the dielectric grating are indeed reflectance dips (see Figure 3a), while they are reflectance peaks in MIM meta-grating (see Figure 3b) except for the resonances as marked by red dotted circular rings, where the Fano resonances between RA modes and the broadband plasmonic resonances are disturbed to a certain extent by the intrinsic absorption of the gold film. (3) The coupling strengths between RA and the broadband plasmonic mode can be tuned by the incident angles (see Figure 3b). Disturbed by the intrinsic absorption of the gold film, the coupling strengths have no evident changes at small incident angles (i.e., θ=0°~8°), as indicated by weak far-field radiation ability of the hybrid RA resonances. As the incident angle increased to 10°, an enhanced reflectance peak near RA mode can be recognized from the spectrum, implying the enhancement of the coupling strengths between these two modes. Meanwhile, we also note other angle-resolved features in Figure 3a,b at the wavelength range of 600–750 nm, which are marked by blue arrows and light-blue dotted lines. The angle-resolved features for those resonance modes are consistent with those of SPPs, thus further verifying their identities.

### 2.2. High-Contrast Fano Resonance and Its Application in Reflective Filters

Continuing to increase the angles of incident light, we observe gradually enhanced Fano couplings between RA and the broadband LSPR. Figure 4a shows the corresponding angle-resolved reflectance spectra of the MIM meta-grating depicted in Figure 1e, where the meta-grating is labeled as “Samp. A” and the incident angles are turned from 10° to 26°. In the angle-resolved spectra, we observe a series of narrowband and high-contrast reflection peaks shifted from 567 nm to 667 nm with relatively high reflectance (75–88%). This implies that the coupling strength between RA and the broadband LSPR is gradually enhanced. The reflectance reaches the peak as the RA mode shifts to the central wavelength of LSPR (i.e., λ = 610 nm) at the incident angle of θ=18°.

Figure 4b shows the measured and simulated reflectance spectra of Samp. A at θ=18°, which are plotted by red circles and blue solid curves, respectively. The reflectance of the hybrid RA mode is up to 88%. Due to strong Fano coupling between RA and the broadband LSPR, the linewidth of the hybrid RA mode in the FWHM is increased to about 20 nm. The black solid curve in Figure 4b shows the CMT fitting spectrum with fitting parameters as follows: radiation damping of RA and broadband LSPR: *Γ_RAe_* = 10, *Γ_LSPRe_* = 25; intrinsic damping of these two modes: *Γ_RAi_* = 0.5, *Γ_LSPRi_* = 55; coupling strength between these two resonance modes: *t* = 28. The fitting result shows good agreement with that of measurement except for the position of intrinsic absorption of the gold film. Obviously, stronger coupling between these two modes enhances the radiation ability of RA mode to a certain degree (*Γ_RAe_* turning from 5 to 10 as the incident angles increased from 0° to 18°), while it also increases the linewidth of the hybrid mode correspondingly, which is actually not appreciated in biological/chemical sensing applications.

The peak location of the hybrid RA can be tuned by changing the central wavelength of LSPR, which can be verified by the angle-resolved reflectance spectra of the MIM meta-grating labeled as “Samp. B”, as shown in Figure 4c. Compared to “Samp. A”, the thickness of the photoresist spacer *d*_2_ and the modulation depth of the PR grating *h* in “Samp. B” are changed to 55 nm and 300 nm, respectively, while other geometric parameters remain unchanged. The central wavelength of LSPR in “Samp. B” is about 630 nm. Correspondingly, the reflectance reaches about 100% near the central wavelength of LSPR at the incident angles of 19°, 21° and 23°. Obviously, the hybrid RA in “Samp. B” exhibits higher reflectance (i.e., 76–100%). Such superior spectral performances of the MIM meta-grating make it promising for the application of high-efficiency and narrowband reflective filters. The diffracted wavelengths of the hybrid RA mode in Figure 4a,c are basically consistent with the theoretical calculation ($\lambda_{RA}=P\left(\sin\left(\theta \right)+n\right)$); they are demonstrated in Figure 4d and plotted by red (Samp. A) or blue (Samp. B) circles (measurement results) and a black solid line (theoretical calculation). 

### 2.3. Sensor Performances of the MIM Meta-Grating

Benefiting from the extremely sharp and asymmetric lineshape of RA diffraction, the FOM of the hybrid sensor based on Fano resonance between RA and the broadband LSPR can be described as follows:(1)$$\mathrm{FOM}=S_{HRA}/\Delta \lambda_{HRA}$$
where the subscript “HRA” denotes the hybrid RA mode. $\Delta \lambda_{HRA}$ represents the linewidth of the hybrid RA mode in the FWHM, which can be optimized by adjusting the incident angles according to the above analysis. $S_{HRA}$ is the sensitivity of the hybrid sensor. It is related both to the detected sample and the electric-intensity distribution of the hybrid RA mode near nanostructures. As for external environment RI detection, the $S_{HRA}$ is determined mainly by that of RA mode ($S_{RA}=P$) because of its extremely high sensitivity to RI changes in a relatively large volume. Correspondingly, the FOM of the hybrid sensor can be described as FOM = $P/\Delta \lambda_{HRA}$. As for local RI detection, such as detecting the bonding of small biological/chemical molecules to the nano-device surface, the sensitivity of the hybrid sensor $S_{HRA}$ should mainly be determined by that of the broadband LSPR for its extremely strong near-field enhancement, smaller mode volume and high sensitivity to local RI change. Therefore, the FOM of the hybrid sensor in this case can be written as FOM = $S_{LSPR}/\Delta \lambda_{HRA}$. Regardless of the kind of sensing application, the overall sensor performance can be optimized by carefully adjusting the incident angles.

To demonstrate the above analysis, we performed these two kinds of sensor simulations. Firstly, we simulated the reflectance spectra of the hybrid sensor immersed in different RI environments (see Appendix A in the Appendix A for the detailed simulation configuration). Figure 5a–c show the simulated reflectance spectra of the hybrid sensor for incident angles of θ = 0°, 10° and 20°, respectively. The environmental RI is changed from 1 to 1.02 with a small step size of 0.005. It should be mentioned that “Samp. A” is applied in this section, while the intrinsic absorption of gold is ignored to avoid its influence on sensor performances. As the incident angle is 0°, a relatively broadband hybrid RA mode due to the degeneracy of ±1 order RA modes appears at about 465 nm in Figure 4a [19]. The sensitivity of the hybrid sensor is about $S_{0{}^{\circ}}\approx \left(474.3-465\right)/0.02\approx 465\;\mathrm{nm}/\mathrm{RIU}$, which shows good agreement with the theoretical expectation. As the incident angles increase to 10° and 20°, the sensitivities of the hybrid sensor are about $S_{10{}^{\circ}}\approx \left(555.5-546\right)/0.02\approx 470\;\mathrm{nm}/\mathrm{RIU}$ and $S_{20{}^{\circ}}\approx \left(624-633.2\right)/0.02\approx 460\;\mathrm{nm}/\mathrm{RIU}$, respectively. The slight deviation between simulation results and theoretical calculations (i.e., *S*=P=465 nm/RIU ) may be due to angular dispersion of the reflectance spectra at large incident angles with a relatively broadband source in FDTD simulations. Figure 5d summarizes the simulated results and re-plots the wavelength shifts of sensor signal as a function of the environmental RI at incident angles of *θ* = 0°, 10° and 20°, which are plotted by different-color dotted lines and decorated with different-color circles. It is obvious that the sensitivity of the hybrid sensors at different incident angles is the same, and the spectral shifts of hybrid RA modes as a function of the refractive index changes are almost linear. Those properties are consistent with the theoretical predictions, while the simulated linewidths of the hybrid RA at different incident angles are different. They are 22 nm, 8.3 nm and 10.1 nm at 0°, 10° and 20°, respectively. Correspondingly, the FOMs of those sensors are 22, 57 and 44.5. Therefore, the hybrid sensor performs better overall, as the incident angle is about 10°, which can be clearly distinguished from the spectral response of such sensors at different incident angles. Moreover, as listed in Table 1, if it is compared with some previous MIM meta-gratings [11,34,38,39,40,41,42,43], the sensing performance of our device is still good.

Then, we sought to detect the local RI changes caused by the bonding of small target molecules to the nano-device surface, instead of exchanging the entire embedding medium. In this sensing simulation, a homogeneous coating of small molecules on the surface of the nano-device is arranged as a uniform dielectric film with a certain thickness and RI. Figure 6a–c show the reflectance spectra of the hybrid sensor which is covered by different thicknesses of thin film (equivalent to different layers of biomolecules) with *n* = 1.5 (typical for organic molecules) for incident angles of *θ* = 0°, 9° and 14°, where the thickness of the coating layer is changed from 0 to 8 nm with a step size of 2 nm. Through parameter sweeping on the incident angles, we ultimately chose three typical sensor spectra at *θ* = 0°, 9° and 14°. Obviously, the hybrid sensor worked at *θ* = 9° is really the best. Figure 5d summarizes the simulated results and re-plots the wavelength shifts of sensor signal as a function of the thickness of coating film at incident angles of *θ* = 0°, 9° and 14° with different-color dotted lines and decorated with different-color circles. Meanwhile, for the sake of fairness, we compare the performance of the hybrid sensor with the simple RA-based sensor worked at *θ* = 9° and the broadband plasmonic sensor worked at *θ* = 0°, which are plotted in Figure 6d by different-color dotted lines and decorated with yellow and purple pentagonal shapes, respectively. According to the figure of merit for thin-layer detection (i.e., ${\mathrm{FOM}}_{layer}^{*}=d\lambda /(dl*\Delta \lambda_{RA})$ with dl denoting the thickness of the coating film) introduced by J. Becker et al. [6], the average ${\mathrm{FOM}}_{layer}^{*}$ of the hybrid sensor worked at *θ* = 9°, the simple RA worked at *θ* = 9° and the LSPR-based sensor is about 0.139, 0.024 and 0.0092, respectively. Obviously, the hybrid sensor worked at *θ* = 9° really performs the best.

Furthermore, the simulated results in Figure 6d help us understand some more intriguing and important phenomena. First, the spectral shift depends on the size of the molecules relative to the volume of the resonance mode field penetrating into the medium. Due to the strong electric-field enhancement of the LSPR and hybrid RA mode in the nanoscale, the sensitivity of those sensors decreases slightly with the increase in molecule layers. Therefore, the spectral shifts of those sensors as a function of the thickness of the thin film are nonlinear. Second, the LSPR-based sensor is the most sensitive to the local RI changes caused by the bonding of thin film, while, affected by the broadband lineshape, the ${\mathrm{FOM}}_{layer}^{*}$ of the LSPR sensor is the lowest. The simple RA-based sensor has a narrow linewidth, and it is not sensitive to local RI changes because of the relatively large mode volume and relatively weak photon state density of RA. Finally, due to both the relatively narrow linewidth of the hybrid RA and its ability to modulate the enhanced electric-intensity distribution near the nanostructure, the FOM of the hybrid sensor in the detection of local RI changes is enhanced to 6 and 15 times that of the simple RA- and LSPR-based sensors, respectively.

## 3. Conclusions

We report a dual-function nano-device based on the tunable Fano resonances between localized surface plasmon resonances and the Rayleigh anomaly in an MIM meta-grating. The meta-grating is composed of a silver nanoshell array, an isolation grating mask and a continuous gold film. Both experimental and simulated results show that the coupling behavior between these two modes can be modulated by the angles of incident light in an optimized meta-grating. Strong couplings between LSPR and RA worked at large incident angles form a series of narrowband and high-contrast reflective peaks in an LSPR-based dark background, making the meta-grating promising for application as a high-efficiency reflective filter. A Fano resonance that is well optimized by carefully adjusting the angles of incident light can switch such a nano-device to a high-sensitivity sensor with an FOM larger than 57 and the ability to detect the bonding of molecules to the nanostructure. Simulated results show that the figure of merit of the hybrid sensor in the detection of local RI changes is 6 and 15 times higher than that of simple RA- and LSPR-based sensors, respectively.

## Figures and Tables

**Figure 1 sensors-23-06462-f001:**
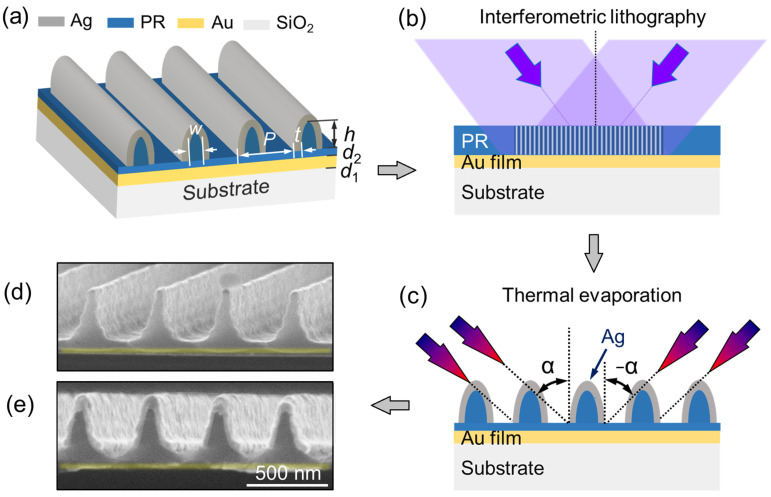
(**a**) Design scheme of the MIM meta-grating composed of an array of silver nanoshells and a continuous gold film separated by a PR grating mask without developing to the end. (**b**,**c**) Fabrication schemes of PR grating mask and the isolated Ag nanoshell arrays, respectively. (**d**,**e**) SEM pictures of part of the fabricated sample before and after the deposition of Ag nanoshells on the surface of the PR grating.

**Figure 2 sensors-23-06462-f002:**
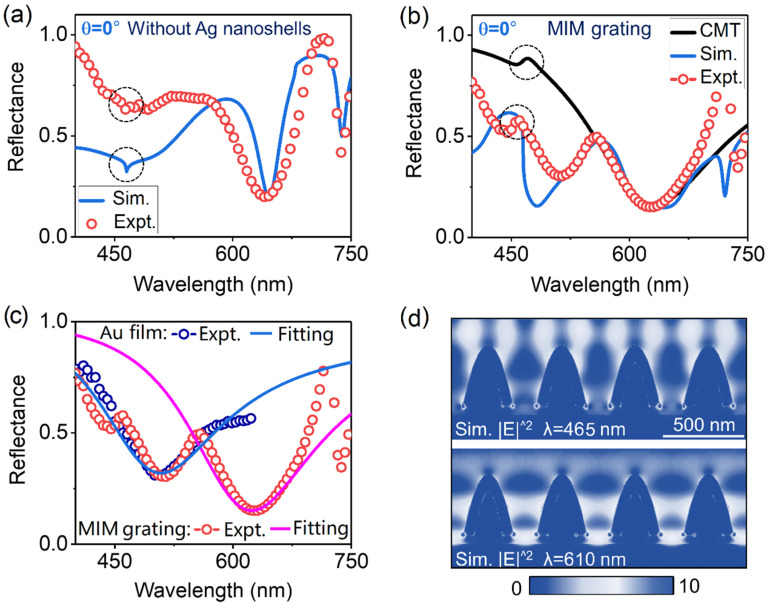
(**a**,**b**) Measured (red circles) and simulated (blue solid curve) reflectance spectra of the samples depicted in Figure 1d,e, respectively, at *θ* = 0°. The black solid curve in Figure 1b denotes the CMT fitting result of the MIM meta-grating. (**c**) Spectral decomposition of MIM meta-grating, where the red circles denote the original measurement spectrum of MIM meta-grating, the pink solid curve denotes the CMT fitting result of the plasmonic mode, the blue circles and blue solid curve represent the measured and CMT fitting spectra of intrinsic absorption of gold, respectively. (**d**) Electric-intensity distribution of the MIM meta-grating at λ ≈ 465 nm and 610 nm, corresponding to the RA and LSPR modes, respectively. The dashed circles in (**a**,**b**) denote the simple RA resonances and hybrid RA resonances, respectively.

**Figure 3 sensors-23-06462-f003:**
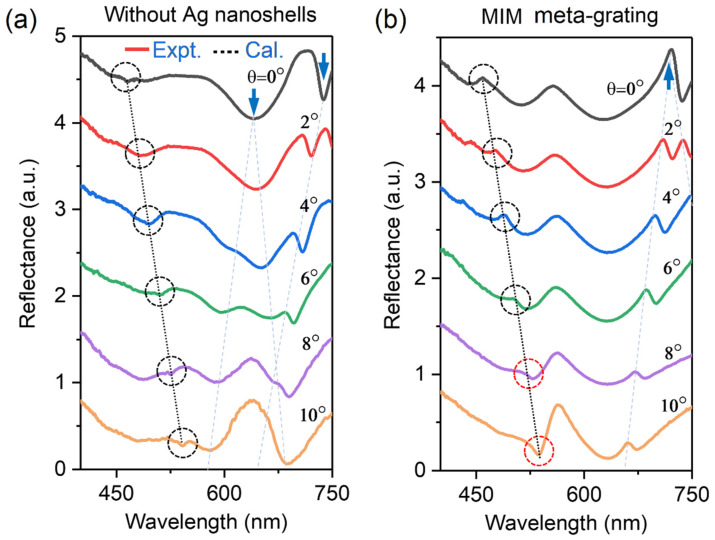
(**a**,**b**) Angle-resolved reflectance spectra of the fabricated samples depicted in Figure 1d,e, respectively, with the incident angles turned from 0° to 10°. The black dotted line denotes the theoretically calculated wavelengths of RA modes. The dashed circles denote the simple RA resonances and hybrid RA resonances, respectively. In this figure, the dashed circles is named as black or red dotted circular rings. The blue arrows denote the SPP modes at a normal incident angle.

**Figure 4 sensors-23-06462-f004:**
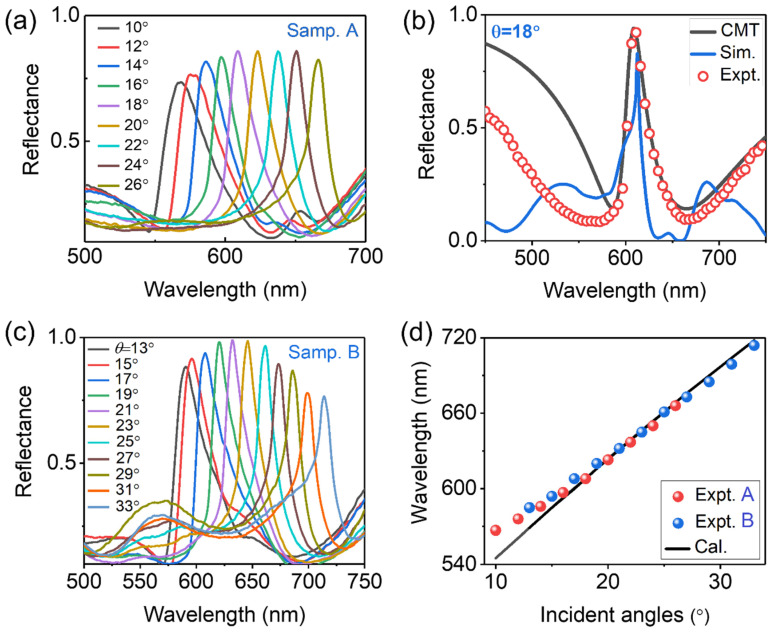
(**a**) Angle-resolved reflectance spectra of the samples depicted in Figure 1e (which is labeled as “Samp. A”) with the incident angles turned from 10° to 26°. (**b**) Measured (red circles), simulated (blue solid curve) and CMT fitting (black solid curve) reflectance spectra of Samp. A at *θ* = 18°. (**c**) Angle-resolved reflectance spectra of “Samp. B” with the incident angles turned from 13° to 33°, with other geometric parameters of “Samp. B” unchanged, except for *d*_2_ and *h* (*d*_2_ = 55 nm and *h* = 300 nm). (**d**) RA diffraction wavelength as a function of incident angles. The red and blue circles correspond to the measurement results of “Samp. A” and “Samp. B”, respectively. The black solid lines represent the theoretical calculations.

**Figure 5 sensors-23-06462-f005:**
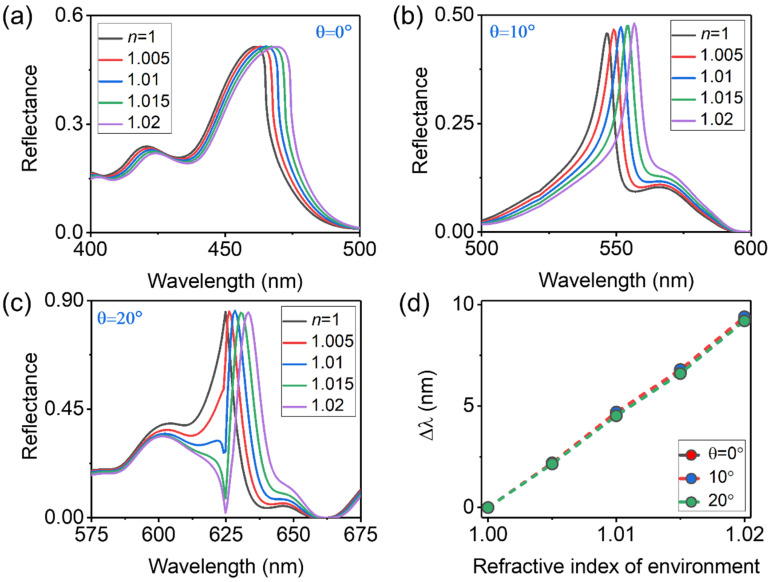
(**a**–**c**) Simulated reflectance spectra of the hybrid sensor immersed in different RI environments changed from 1 to 1.02 for incident angles of *θ* = 0°, 10° and 20°, respectively. (**d**) Wavelength shifts of the hybrid sensor signal as a function of the environmental RI at incident angles of *θ* = 0°, 10° and 20°.

**Figure 6 sensors-23-06462-f006:**
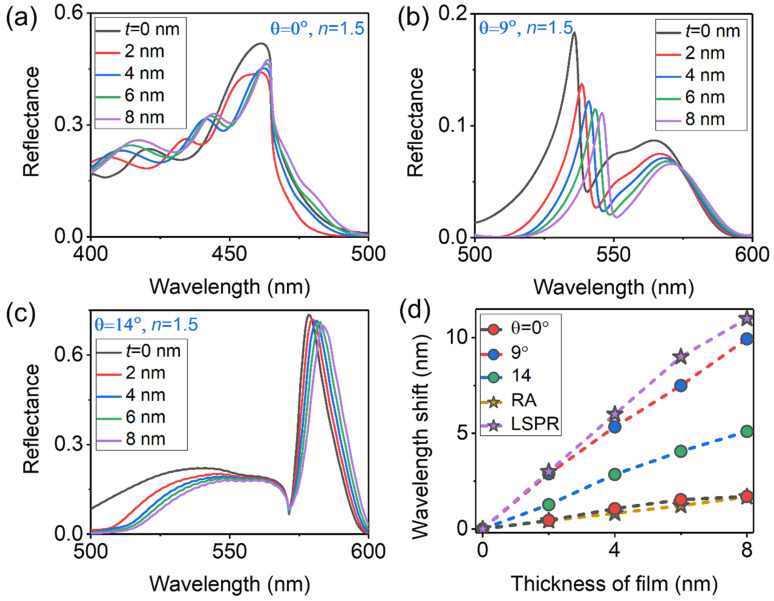
(**a**–**c**) Simulated reflectance spectra of the hybrid sensor which is covered uniformly by different thicknesses of thin film with RI *n* = 1.5 for incident angles of *θ* = 0°, 9° and 14°, respectively. (**d**) Wavelength shift of hybrid sensor signal as a function of the thickness of the coating film at incident angles of *θ* = 0°, 9° and 14°, which are plotted by different-color dotted lines and decorated with different-color circles. The yellow dotted line decorated with yellow pentagonal shapes and the purple dotted line decorated with purple pentagonal shapes represent the performances of simple RA (worked at *θ* = 9°)- and LSPR (at *θ* = 0°)-based sensors, respectively.

**Table 1 sensors-23-06462-t001:** Comparison of sensitivity (S) and the figure of merit (FOM) values in literature.

Ref.	[11]	[34]	[38]	[39]	[40]	[41]	[15]	[42]	[43]	Our Device
Sensitivity (max) nm/RIU	1015	400	2382.5	452	300	1335.69	547.5	695	360	465
FOM (max) (RIU^−1^)	108	2	28.3	30.3	2	6.35	58	96.5	11.2	57
Working wavelength (nm)	1240–1350	1685–1744	2200–8600	500–800	400–700	2000–11,000	600–1000	700–1000	500–900	450–900

## Data Availability

Not available.

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
