# Peer review of "Dual-Function Meta-Grating Based on Tunable Fano Resonance for Reflective Filter and Sensor Applications"

_sensors, 2023, doi:10.3390/s23146462_

Round 1

Reviewer 1 Report

The author proposes a plasmonic sensor based on Fano resonances between surface plasmon resonance and Rayleigh anomaly via a metal-insulator-metal metagrating. The fabrication, simulation, and optimization of metagrating have been presented in the manuscript. However, the theory of surface plasmon resonance and Rayleigh anomaly is not discussed clearly in the manuscript. Therefore, I recommend that the manuscript be accepted after major revision. My comments are listed as follows:     

1.      Figure 1(a) does not demonstrate the fabrication flowchart, such as the photoresist grating pattern and the silver nanoshell deposition. Each fabrication process should be discussed in detail in the manuscript, including the fabrication instruments.

2.      The color bar of the simulated nearfield in Figure 1(g) should be present on the side of the image.

3.      The theory of resonance surface plasmon resonance and Rayleigh anomaly should be presented to discuss Figure 1(d). For example, the equation of the resonance wavelength shows the function of the geometrical parameters of the metagrating.

4.      The simulated dip at 463 nm in Figure 1(d) is not presented clearly. The simulated spectra should be better presented as lines rather than dash lines.  

5.      The color label of curves in Figures 3(a) and (3c) should be present in the picture.

6.      How does the author get the equation of the sensitivity of the metagrating corresponding to P(1+sin (θ))?

7.      The author presents the performance of the metagrating via simulation in Figure 5 by changing the thickness of the thin film from 0 nm to 8 nm with a constant refractive index of 1.5. However, in the experiment, the refractive index of the films in order of nanoscale is related to their thickness. In other words, the complex coupling between the optical constants and the film thickness occurs, as the thin film’s thickness is far less than the wavelength due to the scaling effect. How did the author verify that the experiment would have the same result as the simulation?

8.      To help the readers have a more comprehensive understanding of the new research on metasurfaces, I suggest supplementing some latest works about biosensors with large refractive-index sensitivities [Photonics Research 10(9),2215-2222, 2022]; fano resonance base on multi-layer metamaterial [Optics Letters 47(22), pp 5781-5784, 2022], terahertz liquid crystal programmable metasurface [Optics Letters 47, no. 7 (2022): 1891-1894], Dual-band multifunctional coding metasurface [Photonics Research 10, no. 2 (2022): 416-425], electrically controllable terahertz metamaterials with large tunabilities and low operating electric fields using electrowetting-on-dielectric cells [Optics Letters 46, no. 23 (2021): 5962-5965] and metalens [Photonics Research 10, no. 4 (2022): 886-895].

Proofreading is need for this manuscript

Author Response

First of all, we would like to deeply thank the reviewer for his or her valuable time and comments on our manuscript. We have carefully considered all comments raised by the reviewer and revised our manuscript (marked as red) accordingly. Below are our detailed replies to the reviewers’ comments. Please see the attachment.

Reviewer 2 Report

The authors studied MIM meta-grating for high reflective filter and high refractive index sensor. MIM meta-grating was constructed by silver nanoshells on a continuous gold film. They revealed narrowband and high-contrast reflective peaks by strong couplings between local surface plasmon resonance mode and Rayleigh anomaly mode. The manuscript seems interesting, however, following issues need to be considered: 

(1) Please describe the fabrication process and the condition more in detail how to fabricate the isolation of silver nanoshells as shown in Fig.1(a). According to the references, it may be fabricated by angled deposition. The fabrication conditions are significant points; how much of the deposition angle, vacuum rate, and the thickness.

(2) Why did you apply silver and gold composites for meta-grating? Usually, same metal of silver or gold is used to the shell and film in terms of plasma frequency. What are the advantages of a silver shell and gold film compared with a silver shell and silver film (or gold shell and gold film)? Gold is stable to chemical media and is useful as a biosensor. Silver exhibits higher plasmon resonance. 

(3) The peak mode is different in Fig. 4(a) and Figs. 4(b), 4(c). You should discuss a reflectance peak appeared between 500 and 600 nm wavelength as Fig.4(a) to compare with the different incident angle 10 and 20 degrees. Therefore, the Figs. 4(a), 4(d) and Figs. 5(a), 5(d) should be revised and discussed in the updated results. 

(4) According to the spectral shift described by Figs. 4 and 5, how to distinguish the spectral shift by refractive index change and the thickness increase of the target sample?

After these revisions, the paper can be accepted for publication.

Author Response

(The authors gave the same response as above.)

Round 2

Reviewer 1 Report

The author has great responses to the reviewer’s comments. I appreciated the authors’ effort in modifying their manuscript. However, there are two comments for the revised manuscript.

1. The comparison of previous MIM meta-grating should be better presented in a table. Such as the FOM, range of detection, sensitivity, and Q-factor

2. The simulated current distribution at the resonance frequency should be better demonstrated.

Considering the two comments above, I would like to accept after minor revision this manuscript. 

Author Response

First of all, we would like to deeply thank the reviewer for his or her valuable time and comments on our manuscript. We have carefully considered all comments raised by the reviewer and revised our manuscript (marked as red words and green background) accordingly. Below are our detailed replies to the reviewer’ comments.

Reviewer 2 Report

Based on the proper revisions by the authors in response to reviewer's comments, I recommend the manuscript for publication in its current form.

Author Response

We would like to deeply thank the reviewer for his or her valuable time and comments on our manuscript.